# The Impact of Glucagon-like Peptide 1 Receptor Agonists on Obstructive Sleep Apnoea: A Scoping Review

**DOI:** 10.3390/pharmacy12010011

**Published:** 2024-01-08

**Authors:** Khang Duy Ricky Le, Kelvin Le, Felicia Foo

**Affiliations:** 1Department of General Surgical Specialties, The Royal Melbourne Hospital, Melbourne, VIC 3010, Australia; 2Department of Surgical Oncology, The Peter MacCallum Cancer Centre, Melbourne, VIC 3000, Australia; 3Geelong Clinical School, Deakin University, Geelong, VIC 3220, Australia; 4Department of Medical Education, Melbourne Medical School, The University of Melbourne, Melbourne, VIC 3052, Australia; 5Melbourne Medical School, The University of Melbourne, Melbourne, VIC 3052, Australia; kelvinle1605@gmail.com; 6Department of Pharmacy, The Royal Melbourne Hospital, Melbourne, VIC 3010, Australia; felicia.foo@mh.org.au

**Keywords:** glucagon-like peptide 1 receptor agonists (GLP-1RA), GLP-1 agonists, semaglutide, liraglutide, exenatide, Ozempic, obstructive sleep apnoea (OSA), sleep disordered breathing, obesity

## Abstract

Background: Obstructive sleep apnoea (OSA) and associated hypopnoea syndromes are chronic conditions of sleep-disordered breathing with significant sequelae if poorly managed, including hypertension, cardiovascular disease, metabolic syndrome and increased mortality. Glucagon-like peptide 1 receptor agonists (GLP-1RA) have recently garnered significant interest as a potential therapeutic, attributed to their durable effects in weight loss and glycaemic control in metabolic syndromes, such as obesity and type 2 diabetes mellitus. This has led to significant investment into companies that produce these medications and divestment from traditional gold standard methods of OSA management such as continuous positive airway pressure machines. Despite these sentiments, the impacts of these medications on OSA outcomes are poorly characterised, with no high-quality evidence at this stage to support this hypothesis. This scoping review therefore aims to address the research question of whether GLP-1RAs lead to a direct improvement in OSA and associated hypopnoea syndromes. Methods: A scoping review was performed following a computer-assisted search of Medline, Embase and Cochrane Central databases. Papers that evaluated the use of GLP-1RA medications related to sleep-disordered breathing, OSA or other sleep-related apnoeic or hypopnoeic syndromes were included. Results: Literature search and evaluation identified 9 articles that were eligible for inclusion. Of these, 1 was a study protocol, 1 was a case report, 1 was an abstract of a randomised controlled trial (RCT), 1 was a non-randomised clinical trial and the remaining 5 were randomised clinical trials of variable rigour. All studies evaluated the outcomes of GLP-1RAs in patients with diagnosed OSA or symptoms suggestive of this condition. Conclusion: This scoping review identified early evidence to suggest that GLP-1RAs may improve OSA as defined by reduction in apnoea-hypopnoea index (AHI). This evidence is however conflicting due to contradicting results demonstrated from other studies. Overall, these medications were tolerated well, with minor gastrointestinal side-effects reported in some cases. Of all included studies, the quality of evidence was low, with short lengths of follow-up to identify durable effects of these medications on OSA outcomes and identify adverse events. More rigorous, RCTs with sufficient length of follow-up are required before consideration of formalising these medications into OSA treatment guidelines, frameworks and policies are warranted.

## 1. Introduction

Obstructive sleep apnoea (OSA) is a chronic condition of sleep-disordered breathing with important health outcomes including hypertension, atherosclerotic cardiovascular disease, metabolic syndrome, reduced quality of life and premature mortality [1]. OSA is characterised by evidence of complete or partial upper airway closure associated with temporarily reduced or complete cessation of breathing; termed hypopnoea or apnoea respectively [2]. A putative risk factor for OSA is obesity, which predisposes an individual to upper airway collapse and increased work against the higher levels of adipose tissue during ventilation of sleep.

Obesity management has been revolutionised within the past decade, with glucagon-like peptide-1 (GLP-1) receptor agonists (GLP-1RA) such as semaglutide taking centre stage. These medications, initially developed to assist in the glycaemic control of type 2 diabetes mellitus, have shown durable results with respect to weight loss in non-diabetic patients [3]. The downstream effects of this have been drastic, with paradigm shifts in the approaches to obesity management, with further shifts in the expectation from both patients and doctors about the practicality of weight loss within the enabling obesogenic environment. Such outcomes have also extended across multiple sectors, with analysts in the business and finance sector speculating that GLP-1RAs may alter the prevalence of obesity within the population, with the important downstream effect of reducing the burden of OSA. Consequently, in 2023, there has been significant de-investment of shareholders from companies that develop continuous positive airway pressure (CPAP) machines, which are important in the care of patients with OSA.

Notably, the role of GLP-1RAs in obesity has been extensively studied, however, the thesis of whether these drugs can directly impact OSA outcomes however is poorly characterised in the literature. To date, there have only been a few studies that have explored these outcomes directly. This scoping review therefore aims to resolve this research question of whether GLP-1RAs are associated with reduced OSA outcomes, and by extension, other hypopnoea syndromes (collectively OSAHS) by evaluating the current literature. In doing so, this review provides insight into further translational research opportunities in OSA management to improve the outcomes of patients with OSA in an environment where obesity and OSA remain prevalent. 

## 2. Materials and Methods

### 2.1. Search Strategy

A scoping review was conducted in adherence to the Preferred Reporting Items for Systematic Reviews and Meta-Analyses (PRISMA) guidelines and the Arksey and O’Malley framework for scoping reviews. A computer assisted search was performed on Medline and Embase databases on 18 October 2023. The search combined relevant medical subject headings (MeSH) terms and keywords related to GLP-1 agonists, GLP-1 agonist medications (including their on-market trade names), OSA and other sleep disordered breathing issues. Additional articles were hand searched of reference lists of relevant articles where possible. 

### 2.2. Eligibility Criteria

Peer-reviewed, full-text articles available in English language were assessed. Papers that evaluated the role of GLP-1RAs on OSAHS were considered for this review. Papers were excluded if they were not available in full-text or English language, assessed the role of other anti-obesity or anti-diabetic medications on OSAHS and were pre-clinical trials.

### 2.3. Literature Screening and Data Extraction

Screening by title and abstract was completed by two independent investigators (KL, KL). Studies that were eligible based on the aforementioned criteria were selected for full-text analysis. Papers where a decision could not be made during title and abstract screening also progressed to full-text analysis. The same investigators proceeded to subsequently undertake full-text analysis of articles for inclusion in this review. Disagreement during the process was resolved by consensus. A third independent investigator (FF) was consulted for further input and final decision if consensus was not possible. 

### 2.4. Methodological Quality and Risk of Bias Assessment

Methodological quality and risk of bias of included studies were assessed utilising the Newcastle-Ottawa Scale (NOS) by two independent investigators (KL, KL). Disagreements during this process were resolved by consensus. A third independent investigator (FF) was consulted for further input and final decision if consensus was not possible. Quality was classified by total score into the following: low quality (NOS < 5), fair quality (NOS 6–7) and good quality (NOS 8–9).

## 3. Results

### 3.1. Overview of Included Studies

A total of 329 publications were achieved following a computer-assisted search (Figure 1). Following the removal of duplicate publications, 269 articles underwent screening by title and abstract to assess eligibility, resulting in 251 articles being excluded. The remaining 18 articles progressed to full-text analysis. During this process, 1 article was excluded due to wrong intervention, 2 articles were removed due to wrong study design and 6 articles were removed due to wrong outcomes.

A total of 9 articles met eligibility criteria and were included in the analysis of this scoping review [4,5,6,7,8,9,10,11,12]. Of the included studies, 2 were randomised controlled trials, 4 were observational clinical trials, 1 was a case report, 1 was a research abstract and 1 was a research protocol. Included articles are represented in Table 1.

### 3.2. GLP-1RA and Obstructive Sleep Apnoea and Hypopneoa Syndrome (OSAHS)

Seven studies explored the potential of GLP-1RAs in managing OSAHS. 

Garcia de Lucas et al. reported on the therapeutic potential of liraglutide in a 62-year-old male patient with stage-A1 human immunodeficiency virus infection, type 2 diabetes mellitus (T2DM), hyperlipidaemia, obesity, and moderate OSAHS managed with CPAP [4]. In their case report, apnoea-hypopnea index (AHI) determined by polysomnography was used to confirm OSAHS. The authors found after a 24-week course of liraglutide, there was a noticeable decrease in AHI from 27/h to 7.1/h. Of note, concomitant medications during this treatment regime include metformin (1700 mg/day), levemir (30 IU/day) and the patient was advised to follow a 1500-calorie diet with 1 h walking per day. Blackman et al. conducted a 32-week randomised, double-blinded clinical trial investigating the effects of liraglutide in non-diabetic patients (*n* = 276) with obesity and either moderate (AHI = 15–29.9/h) or severe (AHI ≥ 30/h) OSA [5]. Participants were either given liraglutide (*n* = 134; starting dose 0.6 mg/day, incrementing by 0.6 mg per week, maximum dose 3 mg/day) or placebo treatment (*n* = 142) with diet and exercise regimes as an adjunct. The authors revealed greater mean AHI reductions in liraglutide groups (−12.2/h) compared to placebo groups (−6.1/h) (95% confidence interval (CI), −11.0 to −1.2, *p* = 0.0150). Liu et al. investigated the effects of liraglutide on patients with T2DM and OSAHS (*n* = 53) in comparison to hypoglycaemia medication (controlled; *n* = 40) through polysomnography [6]. The authors showed liraglutide treatment significantly reduced AHI of treatment groups (−3.16 ± 3.52/h) compared to placebo (0.5 ± 1.54) (*p* < 0.05). Jiang et al. through a randomised-controlled trial (RCT) explored the effects of liraglutide in patients diagnosed with T2DM and moderate-severe OSA (AHI > 15) currently undertaking CPAP therapy [7]. Participants were allocated either liraglutide (*n* = 44; starting dose 0.6 mg/day, incrementing by 0.6 mg per week, maximum dose 1.8 mg/day) + CPAP or CPAP therapy alone (control; *n* = 45) for 3 months. This study showed that the liraglutide + CPAP treatment reduced AHI (26.1 ± 7.1/h) compared to CPAP alone (31.6 ± 6.9/h) (*p* < 0.05). Similarly, minimum oxygen saturation improved in the liraglutide + CPAP group (83.4 ± 5.8%) compared to CPAP alone (80.4 ± 5.9%) (*p* < 0.05). Amin et al. report in an abstract the conduct of a 4-week randomized controlled trial that investigated the effects of a GLP-1RA (undisclosed) in participants with a moderate OSA (AHI ≥ 15/h) [8]. Participants either received GLP-1RA (*n* = 18; starting dose 0.6 mg/day, incrementing by 0.6 mg per week, maximum dose 1.8 mg/day) or placebo treatment (*n* = 9) alongside standard care. The mean AHI of the treatment group reduced from 50 ± 32 to 38 ± 30 (*p* = 0.0002), where 70% of participants experienced significant reductions by at least 44%. In comparison, placebo groups showed no significant changes. O’Donnell et al. in a recent proof-of-concept 24-week RCT compared the efficacy of CPAP therapy alone (*n* = 11), liraglutide (*n* = 10; starting dose 0.6 mg/day, incrementing by 0.6 mg per week, maximum dose 3 mg/day) and liraglutide + CPAP combination treatment (same dosage; *n* = 9) in obese participants with moderate–severe OSA (AHI > 15/h) [9]. Altogether, all treatments had significant effects on AHI in comparison to baseline measurements (CPAP 48 ± 20 vs. 3 ± 3; liraglutide 54 ± 21 vs. 42 ± 16; combination 48 ± 17 vs. 5 ± 5; all measured per hr, all *p* < 0.05). However, the AHI of liraglutide treatment itself was significantly inferior (42 ± 16/h) compared to CPAP alone (3 ± 3/h) and combination treatment (5 ± 5/h).

Although there are limited studies, there are trials currently investigating the relationship between GLP-1RA and OSAHS management. One study protocol proposed by Sprung et al. investigated and compared the effects of liraglutide treatment (1.8 mg) alone, liraglutide treatment (1.8 mg) + CPAP and CPAP alone on patients (*n* = 132) with newly diagnosed OSA (AHI > 15/h), obesity (Body mass index (BMI) ≥ 30 kg/m^2^) and T2DM (HbA1c > 47 mM/M). Liraglutide dosages will begin at 0.6 mg/day and increment up to 1.8 mg/day by 0.6 mg/week increments [10]. Although having a relatively established safety profile, many studies documented adverse gastrointestinal side effects including nausea, diarrhoea, constipation and vomiting, which resulted in few cases of non-compliance towards liraglutide treatment [5,7,9].

### 3.3. GLP-1RA and Daytime Sleepiness

Overall, two studies have explored the impact of GLP-1RA on daytime sleepiness. Idris et al. conducted a 22-week placebo-controlled single-blinded study to investigate the effect of exenatide on daytime sleepiness in obese patients with T2DM and no diagnosis of OSAHS (*n* = 8) [11]. Exenatide was given bidaily (5 µg for the first four weeks, 10 µg for the remaining weeks). Daytime sleepiness was measured with the Epworth sleepiness score (ESS; subjective) and the OSLER test (objective). The ESS scores were lower when participants followed exenatide treatment (5.7) compared to baseline (12.3) or placebo (11.3) (*p* = 0.003), indicating reduced subjective sleepiness. The OSLER tests revealed that sleep latency (the time to fall asleep) following the OSLER test was greater when participants followed exenatide treatment (37.7 ± 1.7 min) compared to baseline (32.1 ± 1.7 min) or placebo (29.1 ± 1.7 min), indicating reduced objective sleepiness. Finally, Gomez-Peralta et al. showed in a retrospective observational study on 158 obese patients that ESS scores were significantly reduced compared to baseline values in patients treated with liraglutide at both 1-month (baseline: 6.3 ± 4.6 vs. 1 month: 4.9 ± 3.9; *p* < 0.001) and 3-month (baseline: 5.7 ± 4.4 vs. 3 months: 4.2 ± 3.6; *p* < 0.001) intervals [12].

### 3.4. Quality and Risk of Bias Assessment

Quality and risk of bias assessment was performed utilising the Newcastle-Ottawa Scale (Table 2). The scores ranged from 2 to 5. The median NOS score achieved was 4, with an interquartile range of 2. Overall, this represents low quality of included articles.

## 4. Discussion

The advent of GLP-1RAs has led to a noticeable paradigm shift in the way we approach metabolic diseases, in particular T2DM and obesity. There is robust evidence for the role of these medications in durable control of these conditions and this has led to global interest in the repurposing of these medications for similar conditions as well as sequelae of these metabolic conditions. Specifically, GLP-1RAs have been shown to exhibit multisystemic effects on energy utilisation including insulin synthesis and secretion, delayed gastric emptying and attenuation of hypothalamic orexigenic pathways [13]. The outcomes related to these actions include appetite suppression, reduced blood glucose levels and weight loss [13]. Consequently, there has been significant interest from clinicians, investors and other stakeholders concerning medications such as semaglutide and liraglutide in improving outcomes of OSA. Despite this interest, the true effect of these medications on OSA remains theoretical, such as through its link to weight loss in obese patients rather than stemming from rigorous peer-reviewed RCTs. This study is the first to review the current landscape of the literature to assess whether this hypothesis is accurate. Through our comprehensive search, we have found 9 articles that seek to evaluate the effect of GLP-1RAs in OSAHS [4,5,6,7,8,9,10,11,12].

Our scoping review highlights earlier studies that demonstrated an improvement in daytime sleepiness, a key hallmark of OSAHS, with the introduction of GLP-1RAs therapy [11,12]. Despite this, some limitations must be considered in the interpretation of these results. Notably, there is no gold-standard (such as polysomnography) diagnosis of OSAHS in these patients and a short length of follow-up, thereby failing to address the initial research question of whether GLP-1RAs influence OSAHS outcomes. Furthermore, non-blinded subjective scoring through ESS was utilised and can introduce recall bias to the results. Finally, the studies were of small sample sizes and therefore the generalisability and reliability of such findings must be considered. More recent studies have explored the potential therapeutic effect of GLP-1RA in patients with confirmed OSAHS diagnoses prior to study implementation. These studies showed that GLP-1RA treatment had positive effects on OSAHS outcomes, evidenced predominantly through a reduction in AHI [4,5,6,8,9]. Furthermore, GLP-1RAs were shown to have beneficial outcomes as an adjunct therapeutic alongside CPAP therapy. However, the additive improvements of AHI with a combination of GLP-1RA and CPAP strategies in comparison to CPAP treatment alone are contentious [7,9]. Specifically, GLP-1RA interventions are significantly less effective compared to CPAP treatment; the current gold-standard management for OSA [9]. The nature of these results is more rigorous since these studies determined outcomes using the AHI as calculated by polysomnography, a more objective clinical indicator. In addition, a majority of these studies were RCTs that investigated the effect of GLP-1RAs with placebo treatments (on a background of normal management) to minimize external confounding factors. However, key limitations of these studies include the small sample size and shorter time frames for some studies ranging from 4 weeks to 6 months (which may influence the efficacy of GLP-1RA interventions). Finally, an important aspect to appreciate is that although the safety of GLP-1RA is established, there were some discontinuations noted due to adverse gastrointestinal symptoms, specifically due to nausea [5,7,9]. However, these discontinuations represent a minority of cases in each study. In Blackman et al., 19 participants (11% of participants) ceased due to mild to moderate adverse gastrointestinal symptoms, with only 1 reported severe side-effect of cholelithiasis with liraglutide treatment. Comparatively, Jiang et al. and O’Donnell et al. reported only one case of each of the adverse gastrointestinal symptoms from liraglutide treatment resulting in clinical trial withdrawal (2% and 10% of participants respectively). 

The mechanism for therapeutic GLP-1RA effects on OSAHS outcomes is currently unknown. However, the leading posited theory is its property as an anti-obesity drug, especially since obesity is a major risk factor, driver and exacerbator of OSA [14]. In particular, studies have shown that improvements in ESS and AHI due to GLP-1RA interventions are significantly correlated with reductions in weight loss, BMI and waist circumference [5,6,7,12]. However, Amin et al. report no significant correlations between improvements in AHI and BMI changes in patients with BMI above 30, indicating that the effects of GLP-1RA intervention on OSAHS were independent of weight loss [8]. A major limitation of this study is the 4-week duration (the shortest of the clinical studies screened in our search) and limited sample size [13]. Overall, the evidence at current is not sufficient to guide policies, frameworks and funding for the implementation of GLP-1RAs in OSA management. For patients with OSAHS, the use of GLP-1RAs has many unanswered questions that pose additional barriers to implementation. Specifically, current studies are not sufficiently powered, nor follow-up patients for long enough in duration to highlight a durable effect of GLP-1RAs for reducing OSA, nor to adequately characterise side-effects. Given these medications are likely to be required for years to decades, if not a lifetime, adequate follow-up periods in rigorous trials are therefore warranted. Additionally, the issue of concordance with these medications and how they are delivered must be considered. For patients and consumers, the benefit of CPAP lies in its access from the bedroom. GLP-1RAs, which require regular meetings with health professionals for prescriptions, pose a significant barrier to access as well as a further burden to the already stressed health systems we face. It is clear from the low quality of evidence and contradicting results, a call for further rigorous research is necessary to strengthen and better characterise the effect of GLP-1RAs for OSAHS therapy.

The implementation of GLP-1RA therapy for isolated OSA engenders diverse pharmaceutical considerations and challenges. At present, the accessibility of GLP-1RA therapy for OSA in the absence of diabetes or obesity is limited. Prescribers, pharmacists and consumers confront financial and regulatory barriers, necessitating navigation through complex and restrictive access pathways. The prescription of GLP-1RAs for OSA lacks international regulatory authority approval, thus precluding the likelihood of government or insurance subsidies. Three GLP1-RAs-namely, liraglutide (Saxenda^®^), semaglutide (Wegovy^®^, Ozempic^®^), and tirzepatide (Zepbound^®^)-have secured approval from the U.S. Food and Drug Administration for obesity, while the remaining GLP-1RAs available in the market are solely indicated for glycemic control in T2DM. Regulatory authority approvals substantially inform prescribing guidelines, formulary restrictions, and insurance coverage policies, and are currently restricting access to GLP-1RA medications for OSA. Consequently, large out-of-pocket costs, co-payments, or insufficient insurance coverage pose a significant obstruction to obtaining these medications. This challenge is pertinent in nations with publicly funded healthcare systems and medication subsidies. Although liraglutide is the sole GLP-1RA with regulatory approval for obesity in Australia, its exclusion from the Pharmaceutical Benefits Scheme precludes government subsidy, necessitating prescribers and consumers to bear the full cost of the therapy. This implicates pharmaceutical practice, whereby dispensers must be aware of the significant costs to the consumer in the absence of appropriate regulatory approval and indications for subsidy. Specifically, off-label and private prescriptions of liraglutide at the maximum maintenance dose of 3 mg daily incur an approximate monthly cost of AUD 400, a financial burden that can be exceedingly prohibitive. Furthermore, the other two GLP-1RAs available in Australia, dulaglutide and semaglutide, have not obtained Therapeutic Goods Administration approval for the treatment of obesity or OSA; their current indications are confined to glycaemic control in type 2 diabetes mellitus. The intricacies of instituting GLP-1RA therapy are further exacerbated by persistent global shortages stemming from the escalating popularity of this drug class and its unauthorised use for cosmetic weight loss. The unpredictable availability of GLP-1RAs exacerbates the challenges faced by patients in obtaining the medication and maintaining consistent therapy. In light of these pharmaceutical considerations, there is an imperative need for regulatory amendments, expanded indications, and enhanced accessibility to facilitate GLP-1RA therapy as a viable option for OSA management. Multidisciplinary research into the benefit and efficacy of GLP-1RAs is therefore required, with clinicians working alongside pharmacists to evaluate firstly the true efficacy of these medications in OSAHS outcomes with highly powered and rigorous clinical trials, followed by multidisciplinary input into the development of appropriate and accessible guidelines and frameworks for the integration of these medications if deemed appropriate from the evidence. Additionally, education surrounding the use of these medications, their regulation and accessibility is also warranted in the face of limited global supply to ensure judicious stewardship of these medications.

A strength of this scoping review relates to the diverse range of studies, methodologies, objectives and outcomes evaluated as part of the aim to further characterise the research question of whether GLP-1RAs directly influence OSA outcomes. Further, this review utilises a broad and comprehensive search to acquire both qualitative and quantitative data related to these outcomes of interest. Limitations of this review relate predominantly to the paucity of data in this field, with a vast majority of studies or poor methodological quality. This perhaps is explained by the fact that OSA is a chronic condition and in the context of new GLP-1RA drugs, the effect of the latter will take more time to fully characterise. Therefore, at this time, current evidence should be considered with caution and further rigorous RCTs with adequate duration of follow-up are required to determine the effect of these medications on OSA outcomes. 

## 5. Conclusions

The prospect of a novel, accessible and highly efficacious agent to manage sleep disordered breathing conditions like OSA is highly appealing. There has been a shift to consideration of GLP-1RAs for this purpose, given their durable effects of weight loss and glycaemic control in metabolic conditions intimately related to OSAHS disorders including obesity and diabetes mellitus respectively. Despite these sentiments, our scoping review has only identified 9 studies of low-quality evidence with poor consensus in confirming that GLP-1RAs improve OSAHS outcomes in patients. Furthermore, the mechanisms that derive this effect remain poorly characterised and therefore at this point in time, the consideration of integrating GLP1-RAs into the treatment regime for OSAHS remains in its infancy. This review highlights the need for more rigorous and robust evidence before appropriate evidence-based policies, frameworks and guidelines can be developed to inform the use of these medication in OSA. 

## Figures and Tables

**Figure 1 pharmacy-12-00011-f001:**
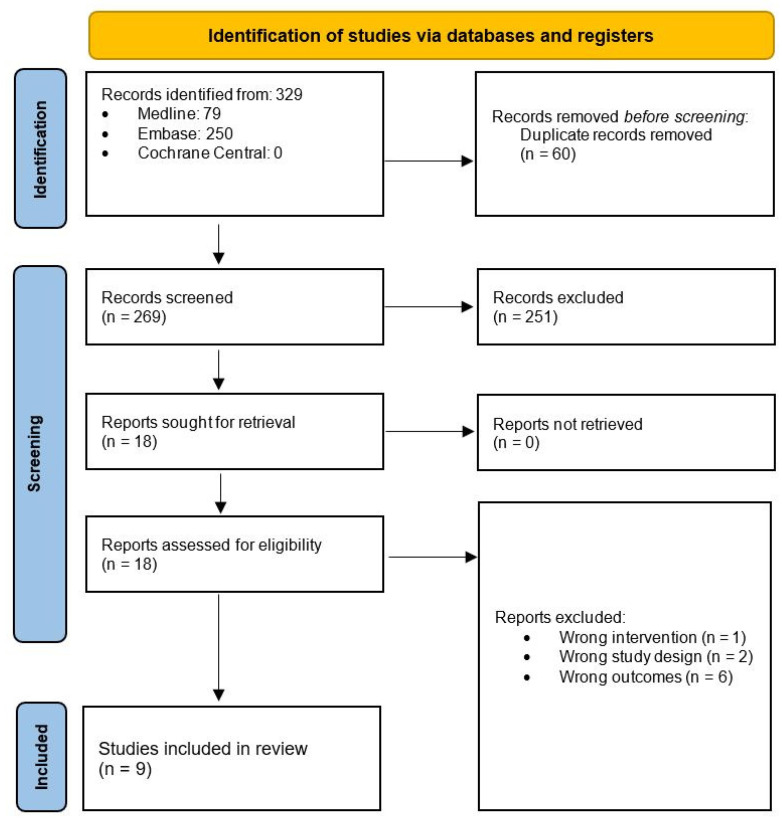
Search strategy and workflow reported in accordance with the PRISMA guidelines.

**Table 1 pharmacy-12-00011-t001:** Overview of included studies.

Author	Year	Study Type	Location	Intervention	Cohort	Level of Evidence (Oxford Centre for Evidence-Based Medicine; OCEBM)
García de Lucas [4]	2015	Case report	Spain	Off-label liraglutide commencing at 0.6 mg/day and uptitrated to 1.2 mg/day	62-year-old man with multiple comorbidities including Human immunodeficiency virus (HIV), type 2 diabetes mellitus, hyperlipidaemia, obesity and moderate sleep apnoea	4
Blackman [5]	2016	Randomised controlled trial	Canada, USA	32 weeks of liraglutide 3.0 mg compared to placebo in addition to diet and exercise	Obese adults aged between 18–64 with diagnosis of moderate or severe obstructive sleep apnoea and unwilling or unable to use continuous positive airway pressure (CPAP) devices	1b
Liu [6]	2020	Retrospective non randomised clinical trial	China	Liraglutide compared to conventional anti-hyperglycaemic drugs	Adults with type 2 diabetes mellitus and obstructive sleep apnoea	2b
Jiang [7]	2022	Randomised controlled trial	China	Off-label liraglutide commencing at 0.6 mg/day and uptitrated to 1.8 mg/day	Adults aged 18–75 with diagnosis of type 2 diabetes mellitus not being treated with dipeptidyl peptidase IV (DPP-IV) or GLP1-RA treatment and severe obstructive sleep apnoea currently being managed with CPAP.	1b
Amin [8]	2015	Abstract of randomised controlled trial	USA	Undisclosed GLP-1RA starting at 0.6 mg/day and uptitrating to 1.8 ng/day	Adults with moderate obstructive sleep apnoea	5
O’Donnell [9]	2023	Randomised proof of concept study	Ireland	Off-label liraglutide regimen alone compared to liraglutide and CPAP compared to CPAP alone.	Non-diabetic patients with moderate to severe obstructive sleep apnoea	2b
Sprung [10]	2020	Protocol	UK	Liraglutide 1.8 mg/day compared to liraglutide and CPAP compared to CPAP alone	Patients with newly diagnosed obstructive sleep apnoea, obesity and type 2 diabetes mellitus	5
Idris [11]	2013	Exploratory placebo controlled clinical trial	UK	Exenatide therapy was initiated at a 5μg twice-daily dose by subcutaneous injection and increased to 10 μg twice daily within 4 weeks of treatment initiation.	Obese adult patients with diabetes but no formal diagnosis of obstructive sleep apnoea	3b
Gomez-Peralta [12]	2015	Retrospective non randomised clinical trial	Spain	Liraglutide	Obese adult patients with type 2 diabetes	2b

**Table 2 pharmacy-12-00011-t002:** Quality and risk of bias assessment of included studies utilising the Newcastle-Ottawa Scale.

Study	Representative of the Exposed Cohort	Selection of External Control/Non-Exposed Cohort	Ascertainment of Exposure	Outcome of Interest not Present at the Start of the Study	Study Controls for Intervention of GLP-1RA	Study Control for External Confounders	Assessment of Outcomes/Ascertainment of Exposure	Sufficient Follow-Up/Same Method of Ascertainment for Cases and Controls	Adequacy of Follow-Up/Non-Response Rate	Total Score (/9)
García de Lucas 2015 [4]	−	−	+	−	−	−	+	−	−	2
Blackman 2016 [5]	−	+	+	−	+	+	+	−	−	5
Liu 2020 [6]	−	−	+	−	+	−	+	−	−	3
Jiang 2022 [7]	−	+	+	−	+	+	+	−	−	5
Amin 2015 [8]	−	−	+	−	+	−	+	−	−	3
O’Donnell 2023 [9]	−	+	+	−	+	+	+	−	−	5
Sprung 2020 [10]	−	−	+	−	+	−	+	−	−	3
Idris 2013 [11]	−	−	+	+	−	+	+	−	−	4
Gomez-Peralta 2015 [12]	−	+	+	+	−	−	+	−	−	4

## Data Availability

No new data was generated or developed in the work related to this manuscript. All referenced data is publicly available on medical databases as described in our methodology. Data can be regenerated but replicating the search of our methodology, which can be found in Appendix A.

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
