# Peer review of "The Impact of Glucagon-like Peptide 1 Receptor Agonists on Obstructive Sleep Apnoea: A Scoping Review"

_pharmacy, 2024, doi:10.3390/pharmacy12010011_

Round 1
Reviewer 1 Report
Comments and Suggestions for Authors
The authors perform a comprehensive review of the literature in the field of GLP1-RA use in sleep apnea patients. This paper is well written and easy to follow. However, it includes only 9 papers from which 1 is an abstract and other a case report. Unfortunately, this is not sufficient to perform a review. Nevertheless, this manuscript would be a great opinion article.
Author Response
The authors perform a comprehensive review of the literature in the field of GLP1-RA use in sleep apnea patients. This paper is well written and easy to follow. However, it includes only 9 papers from which 1 is an abstract and other a case report. Unfortunately, this is not sufficient to perform a review. Nevertheless, this manuscript would be a great opinion article.
We thank the reviewer for their unbiased, constructive and comprehensive review of our manuscript. We believe a scoping review is the appropriate format for this research and concurrent reviewers of this article have also agreed with the methodological robustness of this paper within the realms of a scoping review. This is because as the reviewer has suggested, there is insufficient data, research and evidence to perform a systematic review (and meta-analysis) at this point in time. Given the paucity of data, a scoping review, as we have performed, is appropriate to highlight/evaluate the current landscape of evidence, identify current insights and most important avenues for future directions. The research design of this manuscript therefore remains unchanged.
Reviewer 2 Report
Comments and Suggestions for Authors
This is an important review on an equally important pathology. The review is clear, informative and easy to read. The illustration makes the reading process easier. The conclusions are clear cut and favour a new high quality, more robust, investigation.
Author Response
This is an important review on an equally important pathology. The review is clear, informative and easy to read. The illustration makes the reading process easier. The conclusions are clear cut and favour a new high quality, more robust, investigation.
We thank the reviewer for their in-depth review and feedback related to our manuscript. We agree that there is an important gap within the literature that exists within the space of obstructive sleep apnoea and related syndromes and whether glucagon-like peptide 1 receptor agonists may play a role in revolutionising the management of this pathology. We are therefore highly excited to play a part in the research landscape for this important topic and thank the reviewer in their integral role in evaluating this research.
Reviewer 3 Report
Comments and Suggestions for Authors
The authors present an interesting analysis regarding the impact of GLP-1 analogues and OSA. Please find bellow afew comments for your consideration:
· I recommend removing the brand name from all text, including keywords
· The mechanism of action of GLP-1 agonist should be detailed.
· A list of keywords used for searching should be provided in a manuscript or as a supplementary file
· All abbreviations should be defined (eg. OCEBM, BMI, etc.)
· To a better understanding, the outcomes of the studies should be presented in Table 1 or in another table.
· Tables 1 and 2 must be rearranged
· The References section should be updated with many other titles.
· The references must be set according to MDPI recommendations
Author Response
I recommend removing the brand name from all text, including keywords
We thank the reviewer for their comprehensive and constructive feedback for this paper. We have removed related brand names within the introduction which we believe will improve the balance and remove any concerns relating to conflicts of interest. Importantly, brand names such as Saxenda, Wegovy/Ozempic and ZepBound are still mentioned in the discussion. This we believe is highly relevant to the current landscape of GLP-1 RA management as it discusses the drugs available on the markert, their regulation and market. The discussion is unbiased and we believe should remain in this manuscript to ensure the breadth and quality of research remains.
The mechanism of action of GLP-1 agonist should be detailed.
We thank the reviewer for their comprehensive and detailed feedback of this manuscript. To improve the quality and breadth of this manuscript, we have added a few comments on the mechanism of action of GLP-1RAs into the discussion.
A list of keywords used for searching should be provided in a manuscript or as a supplementary file
We thank the reviewer for their insightful comments. We would like to point out a list of keywords is already included within this manuscript. It can be found between the abstract section and the introduction section.
All abbreviations should be defined (eg. OCEBM, BMI, etc.)
We thank the reviewer for their insight. We have incorporated additional definitions of abbreviations to improve the understanding of this paper.
To a better understanding, the outcomes of the studies should be presented in Table 1 or in another table.
Tables 1 and 2 must be rearranged
We thank the reviewer for their insight and comprehensive feedback. Our opinion is that for all scoping (as for this review) and systematic reviews (in general), it is most important to provide an overview of studies prior to any further tables that analyse these included papers. In this way, an overview of studies (as per table 1) followed by quality assessment of included studies (as per table 2) is a logical progression of research data and therefore no changes have occurred in revision of this.
The References section should be updated with many other titles.
We thank the reviewer for their insight. In the nature of a scoping review, given the paucity of evidence, we have included all the relevant references deemed necessary in this foundational research. It would be inappropriate of us to cite further papers with no clear indication and therefore no changes have been made to this manuscript.
The references must be set according to MDPI recommendations
We thank the reviewer for their in-depth look into this manuscript. We have updated the reference list to ensure it better meets the expectation of the Pharmacy Journal.
Reviewer 4 Report
Comments and Suggestions for Authors
This paper performed a scoping review and a computer assisted search to seek the correlation between the GLP-1RA and OSA. It is well organized and comprehensively described. The topic is original in the field with limited research that could be found. If possible, the author should address some discuss about the possible underlying mechanism that could relate to the GLP-1AR effect on OSA. By now, the conclusions are consistent with the evidence and arguments because there are few papers published in this field. All the references are appropriate. It could be accepted with minor revision in the discussion part. It is well organized and comprehensively described. However , the conclusion is not clear based on limited literature. It is difficult to draw the relationship of one drug to another area of usage when the underlying mechanism for effects on OSAHS is unknown. It's too early to make a conclusion with limited information right now and worth doing further research on this.
Author Response
This paper performed a scoping review and a computer assisted search to seek the correlation between the GLP-1RA and OSA. It is well organized and comprehensively described. The topic is original in the field with limited research that could be found. If possible, the author should address some discuss about the possible underlying mechanism that could relate to the GLP-1AR effect on OSA. By now, the conclusions are consistent with the evidence and arguments because there are few papers published in this field. All the references are appropriate. It could be accepted with minor revision in the discussion part. It is well organized and comprehensively described. However , the conclusion is not clear based on limited literature. It is difficult to draw the relationship of one drug to another area of usage when the underlying mechanism for effects on OSAHS is unknown. It's too early to make a conclusion with limited information right now and worth doing further research on this.
We thank the reviewer for their comprehensive feedback and insight related to this manuscript. We have strengthened the conclusion to further highlight the early landscape of this research at this moment and the need for more robust research.